# A Systematic Review with a Meta-Analysis of the Motivational Climate and Hedonic Well-Being Constructs: The Importance of the Athlete Level

Marc Lochbaum [1,2,*] and Cassandra Sisneros [1]

1 Department of Kinesiology and Sport Management, Texas Tech University, Lubbock, TX 79409, USA; cassandra.sisneros@ttu.edu
2 Research Institute, Education Academy, Vytautas Magnus University, 44248 Kaunas, Lithuania
* Correspondence: marc.lochbaum@ttu.edu

**Abstract:** Motivational climate is known to relate to individual behaviors, emotions, and thoughts. Hedonic or subjective well-being includes self-assessed positive affect (i.e., pleasant affect, moods, and emotions), negative affect (i.e., unpleasant affect, moods, and emotions), and life or domain-specific satisfaction. The aim of this review was to quantify the relationships between task and ego motivational climate scales and measures representing hedonic well-being with sports participants. Potential moderators of the motivational climate and hedonic well-being were examined. This review followed the PRISMA guidelines (PROSPERO ID CRD42023470462, registered 28 October 2023). From five relevant databases, one relevant review, and hand searching, 82 articles totaling 26,378 participants (46.3% female) met the inclusion criteria. The articles spanned publication dates from 1993 to 2023, representing 18 countries, various team and individual sports, and athletes competing in elite (e.g., Olympic) to grassroot (e.g., club sport) competitions. To meta-analyze the motivational climate and hedonic well-being relationships, the random-effects model was used. For the moderation analyses, the mixed-effects model was used. The task or mastery climate relationships were medium in magnitude with positive affect and satisfaction and small with negative affect. The ego or performance climate relationships were small in magnitude for positive affect, negative affect, and satisfaction. Evidence of bias existed in the motivational climate and hedonic well-being relationships. For moderation analyses, athlete level (i.e., elite vs. non-elite) moderated ($p < 0.05$) the task (elite, $r = 0.23$; non-elite, $r = 0.34$) and ego motivational climate (elite, $r = -0.02$; non-elite, $r = -0.13$) and positive affect and satisfaction combined relationships. In conclusion, the motivational climate and hedonic well-being relationships were stronger for the task climate than for the ego climate. The finding that elite athlete correlations appeared dampened is important for future research. Even with the damped relationships, practitioners, from the Olympics to local clubs, should ensure the promotion of the task climate to maximize positive affect and satisfactions in and around the sport experience.

**Keywords:** achievement goal theory; task or mastery climate; ego or performance climate; competitive sports; subjective well-being; quantitative review; elite athletes

## 1. Introduction

Achievement Goal Theory (AGT) research began in the late 1970s, leading to influential publications in the 1980s [1–5]. In the sport and physical activity domains, research has flourished, resulting in quantitative reviews [6–8]. Intertwined with the flourishing of AGT research was great interest in the athletes' motivations, cognitions, continued participation, and well-being as influenced by their coaches, and parents/peers [9,10]. To best study these relationships, sport psychology researchers began creating and validating motivational climate measures with two appearing in 1992, the Perceived Motivational Climate in Sport Questionnaire [11] and the Parent-initiated Motivational Climate Questionnaire [12]. Of the correlates often studied with motivational climate measures [9], no review has

with intention examined hedonic well-being or subjective well-being [13] in the sport context or explored potential moderators. Hedonic or subjective well-being appeared in the literature in the 1980s along with AGT and is a meaningful psychological construct in the human experience comprised of positive and negative affect, and life- or domain-specific satisfaction such as sport. Well-being, hedonic and eudaimonic, within sport and exercise psychology is becoming a popular research topic [14]. Hence, this systematic review aimed to quantify the task and ego motivational climates and measures consistent with the three components of subjective or hedonic well-being relationships.

### 1.1. AGT and Motivational Climate History

AGT is one of the most researched motivation theories across education, psychology, and business. AGT, including both the individual's predisposition and situational influences (e.g., teachers), originated from independent and collaborative research teams in education [1–5]. Via Professor Glyn C. Roberts being part of the early discussions at the University of Illinois, sport researchers began studying and publishing on AGT [15,16]. Since the 1980s, books [17], meta-analyses [6–8,18], and influential articles [19,20] multiplied and thus provide all the relevant information and background of AGT to interested readers. In addition, pertinent to this review, Ntoumanis and Biddle [10] published a review in 1999. More than 20 years later, Lacerda and colleagues provided an extensive review of motivational climate measures in sport [21]. Across sport and physical activity, Harwood et al.'s quantitative review provides a comprehensive listing of motivational climate measures [9]. Thus, we wrote a brief review of Nicholls' AGT and motivational climate measures.

Nicholls [5] built his framework upon the following assumptions: individuals operate in a rational manner and the predominant achievement goal guides the individual's decisions and behaviors in achievement contexts. The demonstration of competence is the goal of action in AGT frameworks. Thus, individual ability perceptions are central to AGT. How individuals reference ability perceptions refer to conceptions of ability. Nicholls theorized ability in two concepts, differentiated and undifferentiated. These ability conceptions define the task or mastery and ego or performance achievement goals, both of which are assumed to be orthogonal and implicit. These two implicit orientations are theorized to determine the vast array of beliefs, emotions, cognitions, and behaviors within achievement settings. Also, both goal orientations reflect ways in which individuals or athletes, the focus of this quantitative review, define success and failure and ways competence is inferred. The task orientation is adopted when personal mastery, achievement of higher ability, and improvement are the prime reasons for motivation. When a task orientation is the focus, athletes define success and failure by self-referenced perceptions of their performance. In contrast to a task orientation, an ego orientation is characterized when an individual's motivation for action is to demonstrate competence, defined by demonstrating superior ability or beating an opponent. Hence, self-comparisons define a task orientation, and other comparisons define an ego orientation.

An athlete's task or mastery or ego or performance involvement is determined by their proneness for each goal state and the current perceived situation [1,2]. As found in Table 1, the PMCSQ, PMCSQ-2, and MCSYS were developed and incorporated into research agendas [9]. The PMCSQ includes two subscales, whereas the second generation of the PMCSQ includes three subscales for each achievement goal orientation. Within their long history of youth sport research at the University of Washington, Smith and Smoll [22] developed a 12-item mastery and ego motivational climate measure. Most recent in the line of motivational climate measures, Appleton and colleagues developed the coach-created Empowering and Disempowering Motivational Climate Questionnaire [23]. This scale includes task- and ego-involving subscales in addition to autonomy-supportive, socially supportive, and controlling coach subscales. For the purpose of this quantitative review, we included all subscales measuring AGT task or mastery and ego or performance subscales.

**Table 1.** The dominant sport motivational climate measures.

| Climate Measure | Subscales | Example Questions |
|---|---|---|
| PMCSQ [11] | Mastery Climate | On this team, trying hard is rewarded. (Mastery Climate) |
| | Performance Climate | The only thing that matters is winning. (Performance Climate) |
| PMCSQ-2 [24] | Task-involving Climate (Subscales: Important Role, Cooperative Learning, and Effort/Improvement) | On this team, the coach wants us to try new skills. (Task-involving) |
| | Ego-involving Climate (Subscales: Intra-Team Member Rivalry, Punishment for Mistakes, and Unequal Recognition) | On this team, the coach makes it clear who they think are the best players. (Ego-involving) |
| MCSYS [22] | Mastery Climate | The coach encouraged us to learn new skills. (Mastery Climate) |
| | Performance Climate | Winning games was the most important thing for the coach. (Performance Climate) |
| EDMCQ-C [23] | Task-involving Climate | |
| | Ego-involving Climate | My coach made sure players felt successful when they improved. (Task-involving Climate) |
| | Autonomy-supportive Climate | |
| | Socially Supportive Climate | My coach yelled at players for messing up. (Ego-involving Climate) |
| | Controlling Climate | |

Abbreviations: PMCSQ = Perceived Motion Climate in Sport Questionnaire [11]; PMCSQ-2 = Perceived Motion Climate in Sport Questionnaire-2 [24]; MCSYS = Motivational Climate Scale for Youth Sport [22]; EDMCQ-C = Empowering and Disempowering Motivational Climate Questionnaire [23].

### 1.2. Study Purposes, Hypotheses, and Research Questions

The two main aims of this quantitative review were to update and extend knowledge of the relationships between the dichotomous motivational climates, task and ego, and hedonic well-being constructs researched in a sport. To our knowledge, only two motivational climate meta-analyses exist in the physical activity domains [9,25] as opposed to motivational climate as one of many correlates (e.g., [7]). Braithwaite and colleagues meta-analyzed motivational climate interventions in PE settings. They reported that task climate interventions improved student self-rated enjoyment ($g = 0.15$) and decreased anxiety ($g = -0.25$) and boredom ($g = -0.27$) along with other outcome variables. Over their 17 categories of correlates, Harwood and colleagues reported upon two main components of hedonic well-being, positive and negative affect. The reported effect size values were consistent with AGT as the task climate was positively related to positive affect ($r = 0.47$) and negatively related to negative affect ($r = -0.17$). In contrast, the ego climate was negatively related to positive affect ($r = -0.11$) and positively related to negative affect ($r = 0.25$). To expand upon Harwood et al.'s review, we searched a broad range of potential positive and negative affect constructs as well as searching for satisfaction. Our secondary aim was to explore potential moderators such as sample makeup (i.e., percent females), athlete level (i.e., elite vs. non-elite), and the sport type (i.e., individual vs. team) of the quantified motivational climates and hedonic well-being relationships.

Based on longstanding AGT proposed and verified relationships, we hypothesized the task climate to correlate positively with positive affect and satisfaction and negatively with negative affect. Conversely, we hypothesized the ego goal climate to be negatively related to positive affect and satisfaction and positively related with negative affect. Regarding our proposed moderators, the motivational climate literature is absent from moderator testing of relationships. Differences between male and female participants within dichotomous AGT studies stem from Duda [19] hypothesizing females to endorse the task orientation more than males and males to endorse the ego orientation more than females. Lochbaum and his colleague [1] found support, with studies using one of the main AGT measures, for males

endorsing the ego orientation more than females. Whether differences in relationships exist, with males potentially being more sensitive to an ego motivational climate, is unknown; yet it is testable. From a series of Norwegian elite athlete research studies, the authors suggested that elite athletes might be more sensitive to their motivational climates [6] and the climate is more influential [26]. Whether this sensitivity or ability to be influenced changes the relationships among the two motivational climates and our hedonic-based correlates is unknown, but worthy of investigation. Concerning sport type, some evidence exists that individual sport athletes are more ego-oriented than team sport athletes [6]. Again, as with our other potential moderators, whether individual athletes are more sensitive to an ego/performance climate, and this sensitivity's impact on the meta-analyzed relationships, is unknown. By using the updated CMA program, our quantitative statistics are more comprehensive than the previous motivational climate quantitative review [9] and thus there is potential for some insights not yet found in the literature such as in addition to our moderator tests.

## 2. Materials and Methods

The Preferred Reporting Items for Systematic Reviews and Meta-Analysis (PRISMA) guidelines [27] guided all aspects of this manuscript. The PRISMA checklist corresponding to this manuscript is found in Supplemental Table S1. For our computations and result interpretations, we utilized Borenstein, Hedges, Higgins, and Rothstein's Comprehensive Meta-Analyses (CMA) Version 4 program and materials [28–30]. The registration information is as follows: PROSPERO ID CRD42023470462, registered 28 October 2023. To avoid self-plagiarism, our methodology and such aspects' subheadings, table titles, and figure captions come from both authors' recent meta-analyses [31–33].

### 2.1. Eligibility Criteria and Selection Process

The inclusion criteria were as follows: (a) a task/mastery or ego/performance motivation climate measure, (b) a hedonic well-being measure, (c) participants engaged in a sport, (d) sufficient data provided for effect size calculation between at least one motivational climate and one hedonic well-being measure, and (e) original data published in a peer-reviewed academic journal. The main exclusion criteria for studies reporting a climate and hedonic well-being measure were as follows: participants sampled in a non-sport setting (e.g., physical education class or leisure non-competitive settings such as exercising at a fitness club) or insufficient data for effect size calculation.

Concerning our search terms, we searched terms within the well-being domain such as flourishing, resilience, burnout, positive affect, negative affect, mental health, depression, satisfaction with life, and satisfaction, with the goal of capturing all relevant studies. Variables such as perceived competence, self-efficacy, confidence, and physical well-being (e.g., injuries) were excluded from the search. Inquiry about missing data or need for clarifications of any kind did not occur. For articles in a language other than English, we used Google Translate, https://translate.google.com/ (accessed on on 1 December 2023), to help find the required data and coding information.

### 2.2. Information Sources, Search Strategy, and Search Protocol

As detailed here and in Figure 1, information sources included references from Harwood et al., databases found within EBSCOhost (search ended 1 November 2023), and hand searching (search ended 1 November 2023). Within EBSCOhost, we selected the following databases: APA PsycArticles, ERIC, Psychology and Behavioral Sciences Collection, PsychINFO, and SPORTDiscus. All search details are outlined below with more details located in Supplemental Table S2.

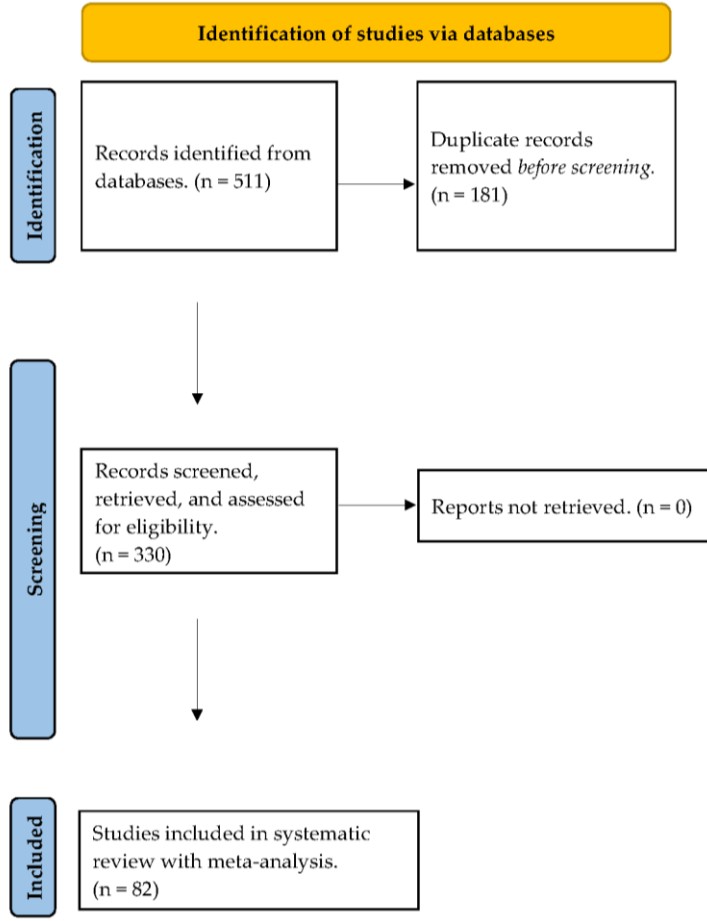

**Figure 1.** PRISMA flow chart for the identification of the included studies.

1.  Examined Harwood et al.'s studies [9] for terms to aid in our search.
2.  Began the EBSCOhost search.
3.  Selected the following individual databases: APA PsycArticles, ERIC, Psychology and Behavioral Sciences Collection, PsychINFO, and SPORTDiscus.
4.  Selected EBSCOhost advanced search.
5.  Typed in search terms using the Boolean operator AND.
6.  Limited EBSCOhost to scholarly peer-reviewed journals.
7.  Selected page options for 50 records per page.
8.  Conducted EBSCOhost searches as described below. After our first search (Search 1), we compared each search to all preceding searches to eliminate duplicates.
9.  Search 1: motivational climate, wellbeing well-being well being, sport ($n$ = 66).
10. Search 2: motivational climate, burnout, sport ($n$ = 25 with 17 non-duplicates).
11. Search 3: motivational climate, flourishing, sport ($n$ = 0).
12. Search 4: motivational climate, resilience, sport ($n$ = 7 with 4 non-duplicates).
13. Search 5: motivational climate, satisfaction with life, sport ($n$ = 8 with 2 non-duplicates).
14. Search 6: motivational climate, depression, sport ($n$ = 7 with 3 non-duplicates).
15. Search 7: motivational climate, positive affect or negative affect, sport ($n$ = 24 with 13 non-duplicates).
16. Search 8: motivational climate, mental health, sport ($n$ = 17 with 10 non-duplicates).
17. Search 9: motivational climate, need satisfaction, sport ($n$ = 31 with 21 non-duplicates).
18. Search 10: motivational climate, satisfaction (in abstract), sport, 1993–1999 ($n$ = 12 with 12 non-duplicates).
19. Search 11: motivational climate, satisfaction (in abstract), sport, 2000–2004 ($n$ = 10 with 8 non-duplicates).

20. Search 12: motivational climate, satisfaction (in abstract), sport, 2005–2009 (*n* = 19 with 12 non-duplicates).
21. Search 13: motivational climate, satisfaction (in abstract), sport, 2010–2014 (*n* = 27 with 6 non-duplicates).
22. Search 14: motivational climate, satisfaction (in abstract), sport, 2015–2017 (*n* = 18 with 10 non-duplicates).
23. Search 15: motivational climate, satisfaction (in abstract), sport, 2018–2019 (*n* = 19 with 8 non-duplicates).
24. Search 16: motivational climate, satisfaction (in abstract), sport, 2020–2021 (*n* = 11 with 5 non-duplicates).
25. Search 17: motivational climate, satisfaction (in abstract), sport, 2022–2023 until 1 November 2023 (*n* = 15 with 5 non-duplicates).
26. Search 18: Review of Birr et al. [34] for studies reporting either the task-involving or ego-involving climate subscales within the EDMCQ-C (*n* = 10 with 8 non-duplicates).
27. Search 19: hand searched Google Scholar with motivational climate AND sport (*n* = 7 non-duplicates).
28. Search 20: motivational climate, anxiety (in abstract), sport 1995–2005 (*n* = 10 with 1 non-duplicate).
29. Search 21: motivational climate, anxiety (in abstract), sport 2006–2010 (*n* = 12 with 0 non-duplicates).
30. Search 22: motivational climate, anxiety (in abstract), sport 2011–2017 (*n* = 27 with 19 non-duplicates).
31. Search 23: motivational climate, anxiety (in abstract), sport 2018–2023 (*n* = 20 with 10 non-duplicates).
32. Search 24: motivational climate, enjoyment or joy or fun or pleasure (in abstract), sport 1992–2000 (*n* = 8 with 6 non-duplicates).
33. Search 25: motivational climate, enjoyment or joy or fun or pleasure (in abstract), sport 2001–2005 (*n* = 9 with 4 non-duplicates).
34. Search 26: motivational climate, enjoyment or joy or fun or pleasure (in abstract), sport 2000–2011 (*n* = 27 with 21 non-duplicates).
35. Search 27: motivational climate, enjoyment or joy or fun or pleasure (in abstract), sport 2012–2015 (*n* = 24 with 19 non-duplicates).
36. Search 28: motivational climate, enjoyment or joy or fun or pleasure (in abstract), sport 2016 (*n* = 8 with 5 non-duplicates).
37. Search 29: motivational climate, enjoyment or joy or fun or pleasure (in abstract), sport 2017 (*n* = 10 with 8 non-duplicates).
38. Search 30: motivational climate, enjoyment or joy or fun or pleasure (in abstract), sport 2018–2019 (*n* = 14 with 8 non-duplicates).
39. Search 31: motivational climate, enjoyment or joy or fun or pleasure (in abstract), sport 2020–2021 (*n* = 9 with 6 non-duplicates).
40. Search 32: motivational climate, enjoyment or joy or fun or pleasure (in abstract), sport 2022–2023 (*n* = 7 with 5 non-duplicates).
41. Search 33: Checked our included studies with Harwood et al. [9] (*n* = 1 non-duplicate).

### 2.3. Data Collection and Items Retrieved

The developed data collection worksheet followed past co-authored systematic reviews and meta-analyses [32,33] with the following data retrieved: climate measurement name, context (sport, PE, or leisure), participant description (e.g., athletes, PE students), correlate data found (yes, no), climate referenced agent other than coach/team (e.g., peers, mother, father), % female participants, participant athletic level description (e.g., Olympic, regional, grassroots, club, university), mean age or age range, sport, apparent country of most participants, well-being measure name, and citation. We used Lochbaum, Cooper, and Limps' [33] classification system (Table 2) to organize the article-published participant descriptions as best as possible.

**Table 2.** Athlete-level categories and specifics used for classification.

| Category | Category Specifics |
|---|---|
| Elite | International competitions at highest level (e.g., Olympics), professional leagues (e.g., Premier leagues), described by authors as elite, samples >18 years of age |
| Advanced | College athletes in all countries, youth/adolescents in country or professional team talent programs, and national-level competition |
| Intermediate | 14–18 years of age, USA high school, club, not identified as elite or in college, in organized training and regional-level competition |
| Recreational | Uni intramural, adults on city teams not listed above at regional level or with extensive training schedules |
| Youth | Sample mean age <14 unless listed in a category above, below high school |
| Mix | Unable to determine one category for sample data |

*2.4. Study Quality and Risk of Bias Assessments*

Table 3 contains the quality questions from Kmet et al. [35]. Both authors rated the studies together with discrepancies discussed until agreement. Based on the question and rating explanations, we eliminated questions 5–7, 9, and 12. Scoring for each question followed Kmet's system of 2, 1, or 0.

**Table 3.** Individual study bias questions and rating explanations summed to a study quality score.

| Quality Questions |
|---|
| 1  Question or objective sufficiently described? |
| 2  Design evident and appropriate to answer study question? |
| 3  Is method of subject selection (and comparison group selection, if applicable) or source of information/input variables (e.g., for decision analysis) described and appropriate? |
| 4  Subject (and comparison group, if applicable) characteristics or input variables/information (e.g., for decision analyses) sufficiently described? |
| 5  If random allocation to treatment group was possible, is it described? N/A: Observational analytic studies. Uncontrolled experimental studies. Surveys. |
| 6  If interventional and blinding of investigators to intervention was possible, is it reported? N/A: Observational analytic studies. Surveys. Descriptive case series/reports. |
| 7  If interventional and blinding of subjects to intervention was possible, is it reported? N/A: Observational studies. Surveys. Descriptive case series/reports. |
| 8  Outcome and (if applicable) exposure measure(s) well defined and robust to measurement/misclassification bias? Means of assessment reported? |
| 9  Sample size appropriate? N/A: Most surveys (except surveys comparing responses between groups or change over time) |
| 10  Analysis described and appropriate? |
| 11  Some estimate of variance (e.g., confidence intervals, standard errors) is reported for the main results/outcomes? |
| 12  Controlled for confounding? N/A: Cross-sectional surveys of a single group. Descriptive studies. |
| 13  Results reported in sufficient detail? |
| 14  Do the results support the conclusions? |

For the risk of bias across studies often referred to as publication bias, we used the following statistics: Orwin's fail-safe *n* [36], the classic fail-safe *n* [37], the funnel plot [38], and Duval and Tweedie's trim and fill [39]. Orwin's fail-safe *n* represents the potential missed studies that would move the correlation past a predetermined threshold. We chose zero as our missed study value and 0.10 or −0.10 as this is the threshold for a small in meaningfulness interpretation. Thus, the greater the value for both fail-safe *n* calculations, the greater the confidence that the result is safe from bias. The classic fail-safe *n* statistic represents the number of null samples required to change a significant value into a non-significant value. We specified the one-tailed test when we conducted the classic fail-safe *n*

analysis. Funnel plots were examined to determine whether the entered studies dispersed in a comparable manner on either side of the overall effect. Symmetry indicates that the retrieved studies captured the essence of all studies. For our last risk of bias across studies metric, we examined Duval and Tweedie's trim and fill analysis. The trim and fill analyses are used to adjust for potential missing studies. Data points filled to the right increase the effect size value, whereas those filled to the left lower the effect size value.

*2.5. Summary Statistics, Planned Analyses, and Certainty Assessment*

The correlation coefficient (*r*) was the summary statistic. The coefficient was based on the random effects model. The random effects model is the logical model given the gathered studies are best thought of as a random sampling of studies published in the literature [30]. Cohen's [40] guidelines of 0.10–0.29 as small, 0.30–0.49 as medium, and 0.50 as large defined meaningfulness. For the most parsimonious and least interrelated summary statistics, we reported only one summary statistic for our six relationships per study. Hence, if a study reported more than one negative affect or mood or the subscales of one measure, those were combined to one effect size. For each overall relationship (e.g., task climate and satisfaction), the number of samples, summary statistic, 95% confidence and prediction intervals, Tau-squared ($\tau^2$) and I-squared ($I^2$), and publication bias statistics were reported. To examine the proposed categorical moderators, a mixed-effects model was used for the calculations. For Orwin's *n*, only the fixed-effects analysis is provided in the CMA program. To assess the potential impact of sample sex makeup, we used a random effects meta-regression model.

To further assess robustness, we conducted the remove-one study and cumulative analyses provided in the CMA program in addition to the classic fail-safe *n* and Orwin's fail-safe *n*, both of which provide statistics indicating robustness. The remove-one study remove-one analysis gauges each study's impact. The remove-one analysis runs the data with all studies except the first, and then all studies except the second, and so on with the resulting data and forest plot depicting the impact of each study. We ran the CMA cumulative analysis program by study publication year. The cumulative analysis run by year allowed us to determine the consistency and thus robustness of the examined relationship over time. Lastly, and in line with the PRISMA guidelines, we examined our results (e.g., 95% confidence and prediction intervals, risk of bias assessments, and differences between moderator groups) to judge certainty related to our hypothesized motivational climate and hedonic well-being relationships.

## 3. Results

*3.1. Study Selection, Characteristics, and Quality*

The 82 included studies are found in Table 4. The 82 studies resulted in 457 extracted correlations entered into the CMA program (see the Supplemental File for all entered correlations). The study publication years ranged from 1993 to 2023, with studies from the following decades: 1990s (*n* = 3), 2000s (*n* = 19), 2010s (*n* = 46), and 2020s (*n* = 14). The studies included 26,378 (M = 321.68, SD = 273.12, range 27 to 1430) participants from Europe (Croatia, Finland, Germany, Greece, Ireland, Italy, Norway, Poland, Portugal, Serbia, Spain, Sweden, Turkey, UK), Asia (China, Japan), and North America (Canada, Mexico, USA). Participants included children, adolescents, and adults, $M_{age}$ = 25.20 (SD = 3.65). Of samples reporting male and female composition, 37% of the samples were greater than 50% female participants (M = 46.20% females). Studies reported on both individual sports athletes (e.g., tennis, swimming, gymnastics) and team sports athletes (e.g., handball, soccer, and volleyball). The levels of competition included elite (*n* = 6), advanced/elite (*n* = 8), advanced (*n* = 15), intermediate (*n* = 20), intermediate/advanced (*n* = 5), mixed (*n* = 13), youth/intermediate (*n* = 4), and youth (*n* = 11) samples. Researchers utilized a variety of motivational climate scales with the most frequently used scales being the PMCSQ-2 (*n* = 42), the PMCSQ (*n* = 21), and the MCSYS (*n* = 12). It is notable that of the included 82 studies, only 20 overlapped with the Harwood et al. [9] quantitative review.

**Table 4.** Study characteristics.

| Study | Year | N (%F) | Country | Level | Sport | Climate | Correlate |
|---|---|---|---|---|---|---|---|
| | | **Participant Characteristics** | | | | **Measures** | |
| Walling et al. [41] | 1993 | 169 (50.8) | US | A/E | Mix Team | PMCSQ | NA, SAT |
| Ntoumanis & Biddle [42] | 1998 | 146 (42.4) | UK | A | Mix Team | PMCSQ | NA |
| Balaguer et al. [43] | 1999 | 219 (33.3) | ES | Mix | Tennis | PMCSQ | SAT |
| Pensgaard & Roberts [44] | 2000 | 69 (28.9) | NO | E | Mix IND, Team | PMCSQ | NA |
| Newton et al. [24] | 2000 | 385 (100) | US | I/A | Volleyball | PMCSQ-2 | PA, NA, SAT |
| Balaguer et al. [45] | 2002 | 181 (100) | ES | A | Handball | PMCSQ-2 | SAT |
| Carr & Wyon [46] | 2003 | 181 (87.2) | UK | A | Dance | PMCSQ-2 | NA |
| Boixadós et al. [47] | 2004 | 472 (0) | ES | Y | Soccer | PMCSQ | PA, SAT |
| Cecchini et al. [48] | 2005 | 82 (0) | **ES** | I/A | Soccer | PMCSQ-2 | PA, NA, SAT |
| Vazou et al. [49] | 2006 | 493 (25.1) | UK | I | Mix IND, Team | PMCSQ-2 | PA, NA |
| Smith, Balaguer et al. [50] | 2006 | 223 (0) | ES | Y | Soccer | PMCSQ-2 | PA, SAT |
| Smith, Smoll et al. [51] | 2006 | 1038 (44.9) | US | Y | Mix Team | MCSYS | NA |
| Cumming et al. [52] | 2007 | 268 (39.1) | US | Y | Basketball | PMCSQ-2 | PA |
| Abrahamsen et al. [53] | 2008a | 190 (46.8) | NO | A/E | Mix IND | PMCSQ | NA |
| Abrahamsen et al. [26] | 2008b | 143 (48.2) | NO | E | Handball | PMCSQ | NA |
| Lemyre et al. [54] | 2008 | 141 (42.5) | NO | E | Mix IND | PMCSQ | SAT |
| Papaioannou et al. [55] | 2008 | 863 (43.1) | GR | Y/I | Mix IND, Team | PSAEGO | SAT |
| Quested & Duda [56] | 2009 | 59 (64.4) | UK | A | Hip Hop Dancing | PMCSQ | PA, NA |
| Steffen et al. [57] | 2009 | 1430 (100) | NO | I | Soccer | PMCSQ | NA |
| Vosloo et al. [58] | 2009 | 151 (61.5) | US | I | Swimming | PMCSQ-2 | NA |
| Weiss et al. [59] | 2009 | 141 (100) | US | I | Soccer | PMCSQ-2 | PA |
| Bortoli et al. [60] | 2009 | 473 (45.8) | IT | Y | Mix IND, Team | PMCSQ | PA, NA |
| Quested & Duda [61] | 2010 | 392 (74.7) | UK | A | Dancing | PMCSQ-2 | PA, NA |
| Holgado et al. [62] | 2010 | 511 (31.1) | ES | E | Mix IND, Team | PMCSQ-2 | SAT |
| Barić [63] | 2011 | 388 (0) | HR | I | Mix Team | PMCSQ | NA |
| O'Rourke et al. [64] | 2011 | 307 (60.2) | US | I | Swimming | PIMCQ-2 | NA |
| MacDonald et al. [65] | 2011 | 510 (52.5) | CA | Mix | Mix IND, Team | MCSYS | PA |
| Núñez et al. [66] | 2011 | 399 (29.5) | **MX** | Mix | Mix IND, Team | PMCSQ-2 | PA |
| Trenz & Zusho [67] | 2011 | 119 (64.7) | US | Mix | Swim | PMCSQ-2 | SAT |
| Bortoli et al. [68] | 2011 | 320 (50.0) | IT | Y | Mix IND, Team | PMCSQ | PA, NA |
| Garcia-Mas et al. [69] | 2011 | 54 (0) | ES | Y | Soccer | MCSYS | NA |
| Bortoli et al. [70] | 2012 | 382 (0) | IT | I | Soccer | PMCSQ | PA, NA |
| Nordin-Bates et al. [71] | 2012 | 327 (75.7) | UK | Mix | Dance | PMCSQ-2 | NA |
| Gillham et al. [72] | 2013 | 396 (57.3) | US | A | Mix IND, Team | MCSYS | SAT |
| Alfermann et al. [73] | 2013 | 56, 117 (51.2) | DE, JP | A/E | Swim | PMCSQ | SAT |
| Eys et al. [74] | 2013 | 997 (53.3) | CA | I | Mix IND, Team | MCSYS | PA |
| Kipp & Weiss [75] | 2013 | 309 (100) | US | I | Gymnastics | MCSYS | PA |
| Atkins et al. [76] | 2013 | 227 (100) | US | Mix | Soccer | PeerMCYSQ | PA |
| Blecharz et al. s1 [77] | 2014 | 56 (64.0) | PL | A/E | Mix Team | PMCSQ-2 | SAT |
| Blecharz et al. s2 [77] | 2014 | 113 (0) | PL | A | Soccer | PMCSQ-2 | SAT |
| Draugelis et al. [78] | 2014 | 182 (86.3) | US | A | Dance | PMCSQ-2 | PA, NA |
| García-Calvo et al. [79] | 2014 | 303 (0) | ES | A | Soccer | PMCSQ-2, PeerMCYSQ | SAT |
| O'Rourke et al. [80] | 2014 | 228 (59.2) | US | I | Swimming | PIMCQ-2 MCSYS | NA |
| Stark & Newton [81] | 2014 | 83 (100) | US | I | Dance | PMCSQ-2 | PA, NA |
| Guzmán & García [82] | 2014 | 303 (82.8) | ES | Mix | Dance | PMCSQ | PA, SAT |
| Solstad & Lemyre [83] | 2014 | 202 (51.0) | NO | Mix | Swim | MCSYS | PA, NA |
| Mora et al. [84] | 2014 | 40 (NR) | **ES** | Y/I | Basketball | PMCSQ-2, PeerMCYSQ | NA |
| Jaakkola et al. [85] | 2015 | 265 (0) | FI | A | Ice Hockey | MCPES | PA |
| Pineda-Espejel et al. [86] | 2015 | 211 (54.7) | **ES** | A | Mix IND, Team | PMCSQ-2 | NA |
| Abrahamsen & Kristiansen [87] | 2015 | 27 (0) | Mix | E | Soccer | PMCSQ | NA |
| Bekiari & Syrmpas [88] | 2015 | 324 (40.1) | GR | I | Mix IND, Team | LAPOPEQ | SAT |
| Weiss [89] | 2015 | 491 (49.6) | US | I/A | NR | PMCSQ-2 PeerMCYSQ | PA |
| Atkins et al. [90] | 2015 | 405 (0) | US | Y | Mix IND, Team | PIMCQ-2 PMCSQ-2 MCSYS | PA |
| Curran et al. [91] | 2015 | 260 (57.6) | UK | Y | Soccer | PMCSQ-2 | PA |
| Abraldes et al. [92] | 2016 | 163 (43.5) | ES | A | Swim | PMCSQ-2 | SAT |
| Bono & Livi [93] | 2016 | 96 (44.0) | **IT** | A/E | Swim | PMCSQ | NA |
| Dorsch et al. [94] | 2016 | 226 (39.8) | US | Mix | Mix Team | MCSYS | PA, NA |
| Tamminen et al. [95] | 2016 | 451 (45.2) | CA | Mix | Mix Team | PeerMCSYS | PA |
| Ruiz et al. [96] | 2017 | 494 (42.7) | FI | A/E | Mix IND, Team | PMCSQ-2 | PA, NA |
| Zanatta et al. [97] | 2018 | 824 (40.6) | FI | I | Mix Team | PMCSQ | PA |
| Al-Yaaribi & Kavussanu [98] | 2018 | 358 (0) | UK | I | Soccer | PMCSQ-2 | PA, NA |
| Monteiro et al. [99] | 2018 | 799 (43.6) | PT | I/A | Swimming | MCSYS | PA |
| Gjesdal et al. [100] | 2018 | 1359 (42.2) | SE | Y | Soccer | PMCSQ-2 | PA, NA |
| Sheehan et al. [101] | 2018a | 38 (47.3) | IE | A | Mix Team | PMCSQ-2 | NA |
| Sheehan et al. [102] | 2018b | 215 (65.0) | IE | A/E | Mix Team | PMCSQ-2 | NA |
| Ruiz et al. [103] | 2019 | 217 (41.9) | FI | A/E | Mix IND, Team | PMCSQ-2 | PA, NA |
| Harwood et al. [104] | 2019 | 92 (35.8) | UK | I | Tennis | MCISCQ-F/M | PA |
| Calvo & Topa [105] | 2019 | 151 (NR) | ES | Mix | Soccer | PMCSQ-2 | SAT |
| Haugen et al. [106] | 2020 | 532 (31.4) | NO | A | Soccer | PMCSQ | SAT |
| Gómez-López et al. [107] | 2020 | 479 (47.8) | ES | I | Handball | PMCSQ-2 | NA |
| Trbojević et al. [108] | 2020 | 117 (100) | RS | Y/I | Mix Team | PMCSQ-2 | SAT |
| Wu et al. [109] | 2021 | 685 (44.9) | CN | A | Mix IND, Team | PMCSQ-2 | NA |
| Pineda-Espejel et al. [110] | 2021 | 217 (48.3) | ES | E | Mix IND, Team | PMCSQ-2 | NA |

**Table 4.** *Cont.*

| | | Participant Characteristics | | | | Measures | |
|---|---|---|---|---|---|---|---|
| Study | Year | N (%F) | Country | Level | Sport | Climate | Correlate |
| Scott et al. [111] | 2021 | 109 (35.8) | US | I/A | Mix Team | PMCSQ | PA |
| Robazza et al. [112] | 2021 | 302 (41.0) | IT | Mix | Mix IND, Team | PMCSQ-2 | PA, NA |
| Morales-Belando et al. [113] | 2021 | 94 (2.1) | ES | Y | Basketball | PMCSQ-2 | PA |
| Trbojević Jocić & Petrović [114] | 2021 | 383 (50.1) | RS | Y/I | Mix Team | PMCSQ-2 | PA |
| Sarı & Bizan [115] | 2022 | 180 (54.4) | TR | I | Mix IND, Team | PIMCQ-2 | PA |
| Santos-Rosa et al. [116] | 2022 | 258 (100) | ES | I | Rhythmic Gymnastics | PMCSQ-2 | PA, NA |
| Robazza et al. [117] | 2022 | 459 (43.7) | IT | Mix | Mix IND, Team | PMCSQ-2 | PA, NA |
| Amaro et al. [118] | 2023 | 109 (0) | PT | I | Soccer | MCSYS PIMCQ-2 | PA |
| Habeeb et al. [119] | 2023 | 150 (43.3) | US | I | Mix IND, Team | PeerMCYSQ MCSYS | PA |

Bold country abbreviation = study written in non-English. Country abbreviations: Canada (CA), China (CN), Croatia (HR), Finland (FI), Spain (ES), Germany (DE), Greece (GR), Ireland (IE), Italy (IT), Japan (JP), Mexico (MX), Norway (NO), Poland (PL), Portugal (PT), Serbia (RS), Sweden (SE), Turkey (TR), and United States of America (US). Level abbreviations: A = advanced, E = elite, I = intermediate, Y = youth. Sport type abbreviations: IND = individual sport. Correlate abbreviations: PA = positive affect correlates—pleasant affect, moods, and emotions, NA = negative affect correlates—unpleasant affect, moods, and emotions, SAT = satisfaction correlates—life and sport domain specific.

Concerning the quality score (see Figure 2 for details), the mean summary score was 0.92 (SD = 0.05) for the rated samples. Though cross-sectional studies are of low quality compared to experimental or quasi-experimental designs, for our purpose of meta-analyzing correlate relationships, the studies were of sufficient quality. The most neglected category was #10 as few studies reported correcting for alphas.

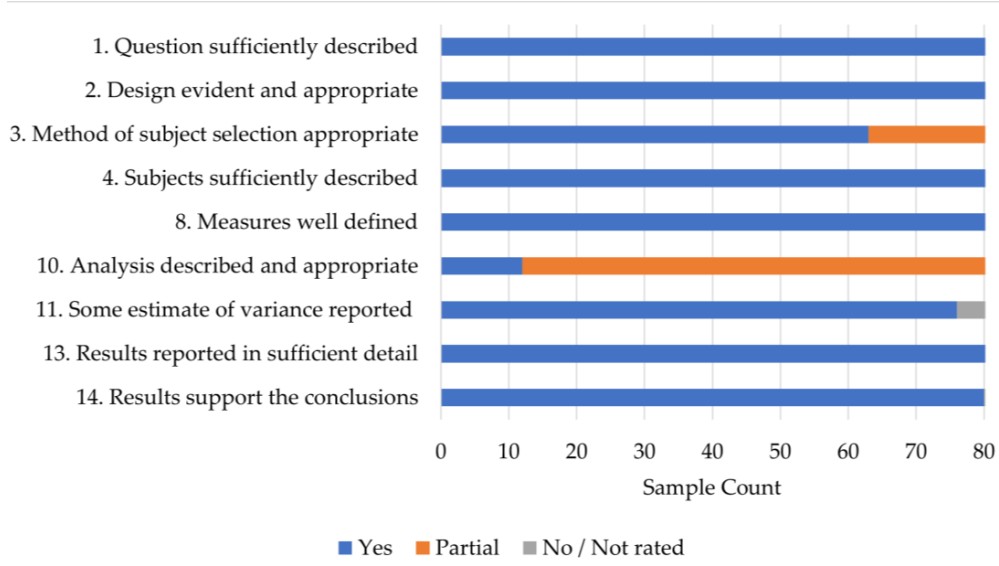

**Figure 2.** Study quality results.

*3.2. Task Climate Individual Study Data, Synthesis of Results, and Risk of Bias across Studies*

Table 5 contains all the summary data for the task climate analyses. The individual study data with corresponding forest plots and the trim and fill plots for the task climate analyses are located in Figures 3–8. For both positive affect and satisfaction, the random effect sizes were medium in magnitude. The 95% confidence intervals remained in the same effect size interpretation range. Of note, the task climate positive affect true prediction interval did not cross zero. Heterogeneity was present though the bias statistics suggested the relationships to be free or mostly free from bias (see funnel plots in Figure 4 for positive affect and Figure 6 for satisfaction).

**Table 5.** Task/mastery climate and hedonic well-being results.

| Correlate | | Effect Size Statistics | | | Heterogeneity Statistics | | | Bias Statistics | | | |
|---|---|---|---|---|---|---|---|---|---|---|---|
| | k | r | 95% CI | 95% PI | Q | $\tau^2$ | $I^2$ | FS | Orwin | Trim/Fill | r [95% CI] |
| PA | 46 | 0.31 | 0.27, 0.35 | 0.04, 0.57 | 662.11 | 0.02 | 93.20 | 43,500 | 89 | 0 | No change |
| NA | 40 | −0.13 | −0.17, −0.08 | −0.36, 0.13 | 451.80 | 0.02 | 91.37 | 4886 | 7 | 7R | −0.09 [−0.13, −0.04] |
| | 39 [A] | −0.13 | −0.18, −0.09 | −0.37, 0.11 | 433.46 | 0.02 | 91.23 | 5067 | 7 | 8R | −0.09 [−0.14, −0.05] |
| Satisfaction | 21 | 0.37 | 0.28, 0.46 | −0.08, 0.70 | 421.18 | 0.05 | 95.24 | 7405 | 49 | 3R | 0.41 [0.32, 0.49] |

Abbreviations: PA = positive affect constructs, NA = negative affect constructs, k = number of samples, CI = confidence interval, PI = prediction interval, Q = Q total between statistics, $\tau^2$ = tau-squared, $I^2$ = ratio of excess dispersion to total dispersion, FS = fail-safe number. Superscript: [A] = Abrahamsen and Kristiansen [87] data point removed.

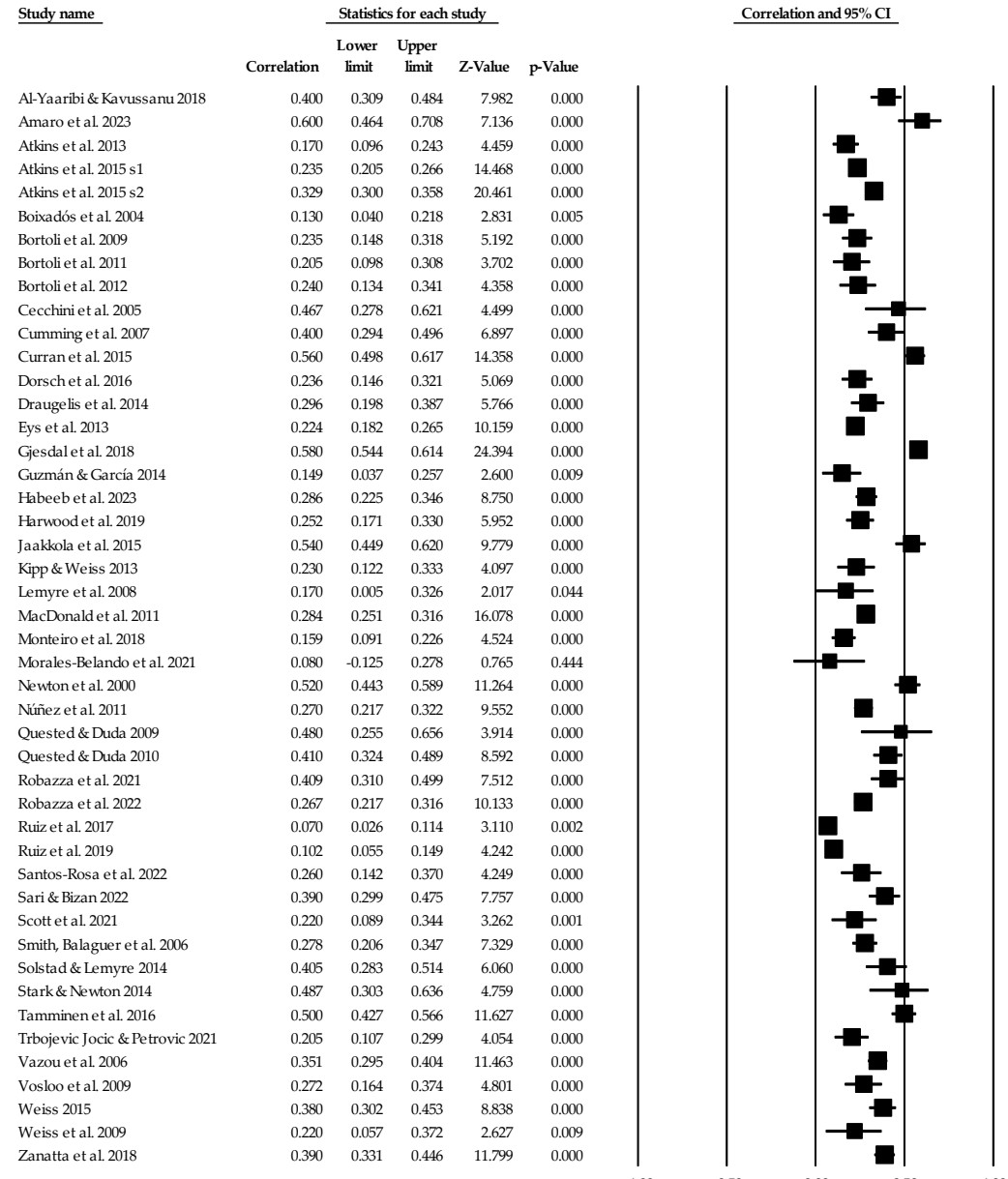

**Figure 3.** Task climate and positive affect statistics expressed as correlations (*r*) with corresponding forest plots. Figure references [24,47–50,52,54,56,58–61,65,66,68,70,74–76,78,81–83,85,89–91,94–99,103, 104,111–120].

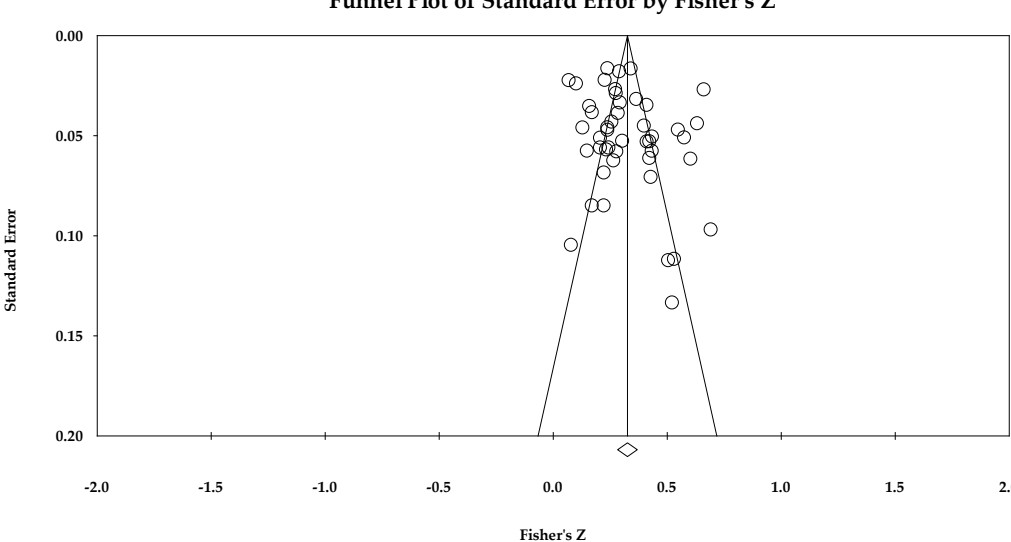

**Figure 4.** Task climate and positive affect random effects plot trimmed and filled. The open circles are the data points. The clear rhombus is the mean effect size.

| Study name | Statistics for each study | | | | | Correlation and 95% CI |
|---|---|---|---|---|---|---|
| | Correlation | Lower limit | Upper limit | Z-Value | p-Value | |
| Abraldes et al. 2016 | 0.203 | 0.096 | 0.306 | 3.686 | 0.000 | |
| Alferman et al. 2013 s1 | 0.080 | -0.187 | 0.336 | 0.584 | 0.559 | |
| Alferman et al. 2013 s2 | 0.350 | 0.180 | 0.500 | 3.902 | 0.000 | |
| Balaguer et al. 1999 | 0.293 | 0.221 | 0.361 | 7.671 | 0.000 | |
| Balaguer et al. 2002 | 0.210 | 0.066 | 0.345 | 2.844 | 0.004 | |
| Bekiari & Syrmpas 2015 | 0.835 | 0.799 | 0.865 | 21.579 | 0.000 | |
| Blecharzet et al. 2014 s1 | 0.620 | 0.427 | 0.759 | 5.278 | 0.000 | |
| Blecharzet et al. 2014 s2 | 0.161 | 0.031 | 0.287 | 2.414 | 0.016 | |
| Boixadós et al. 2004 | 0.560 | 0.495 | 0.619 | 13.705 | 0.000 | |
| Bono & Livi 2016 | 0.283 | 0.087 | 0.458 | 2.806 | 0.005 | |
| Calvo & Topa 2019 | 0.300 | 0.147 | 0.439 | 3.765 | 0.000 | |
| García-Calvo et al. 2014 | 0.116 | 0.060 | 0.171 | 4.036 | 0.000 | |
| Gillham et al. 2013 | 0.263 | 0.197 | 0.327 | 7.545 | 0.000 | |
| Guzmán & García 2014 | 0.342 | 0.239 | 0.438 | 6.172 | 0.000 | |
| Haugen et al. 2020 | 0.271 | 0.190 | 0.348 | 6.393 | 0.000 | |
| Holgado et al. 2010 | 0.185 | 0.100 | 0.267 | 4.218 | 0.000 | |
| Newton et al. 2000 | 0.410 | 0.323 | 0.490 | 8.514 | 0.000 | |
| Papaioannou et al. 2008 | 0.310 | 0.275 | 0.345 | 16.287 | 0.000 | |
| Trbojevic Jocic et al. 2020 | 0.610 | 0.482 | 0.713 | 7.569 | 0.000 | |
| Trenz & Zusho 2011 | 0.600 | 0.471 | 0.704 | 7.465 | 0.000 | |
| Walling et al. 1993 | 0.390 | 0.254 | 0.511 | 5.306 | 0.000 | |

**Figure 5.** Task climate and satisfaction statistics expressed as correlations (*r*) with corresponding forest plots. Figure references [24,41,43,45,47,55,62,67,72,73,77,79,82,88,92,93,105,106,108].

The task climate and negative affect relationship unlike the positive affect/mood and satisfaction relationships was small in magnitude with the 95% confidence intervals crossing 0. As with the positive affect/mood and satisfaction analyses, heterogeneity was present. The trim and fill analysis suggested that bias was present. As seen in the individual study data (see Figure 7) and corresponding funnel plot (see Figure 8), the Abrahamsen and Kristiansen [87] data point appears as an obvious deviation from the other studies. Thus, we examined the task climate and negative affect relationship without Abrahamsen

and Kristiansen. However, these analyses resulted in little to no change in the effect size statistics (refer back to Table 5).

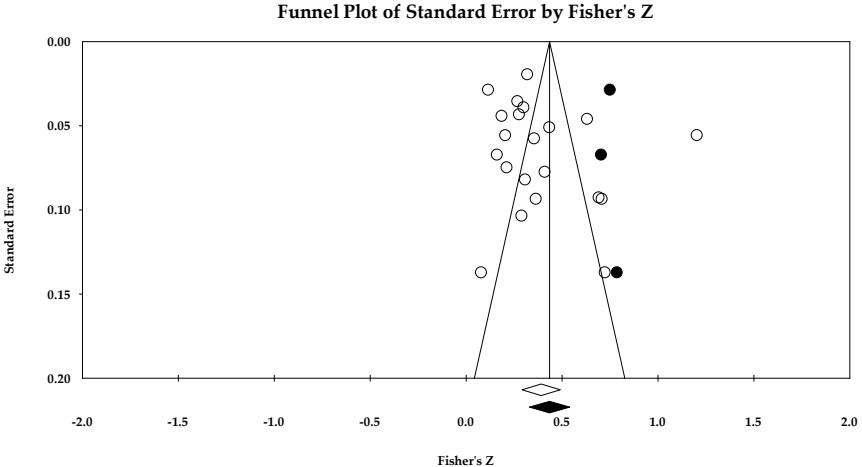

**Figure 6.** Task climate and satisfaction random effects plot trimmed and filled. The open circles are the data points, and the filled circles are the result of the trim and fill analysis. The clear rhombus is the mean effect size, and the filled rhombus is the trim and filled mean effect size.

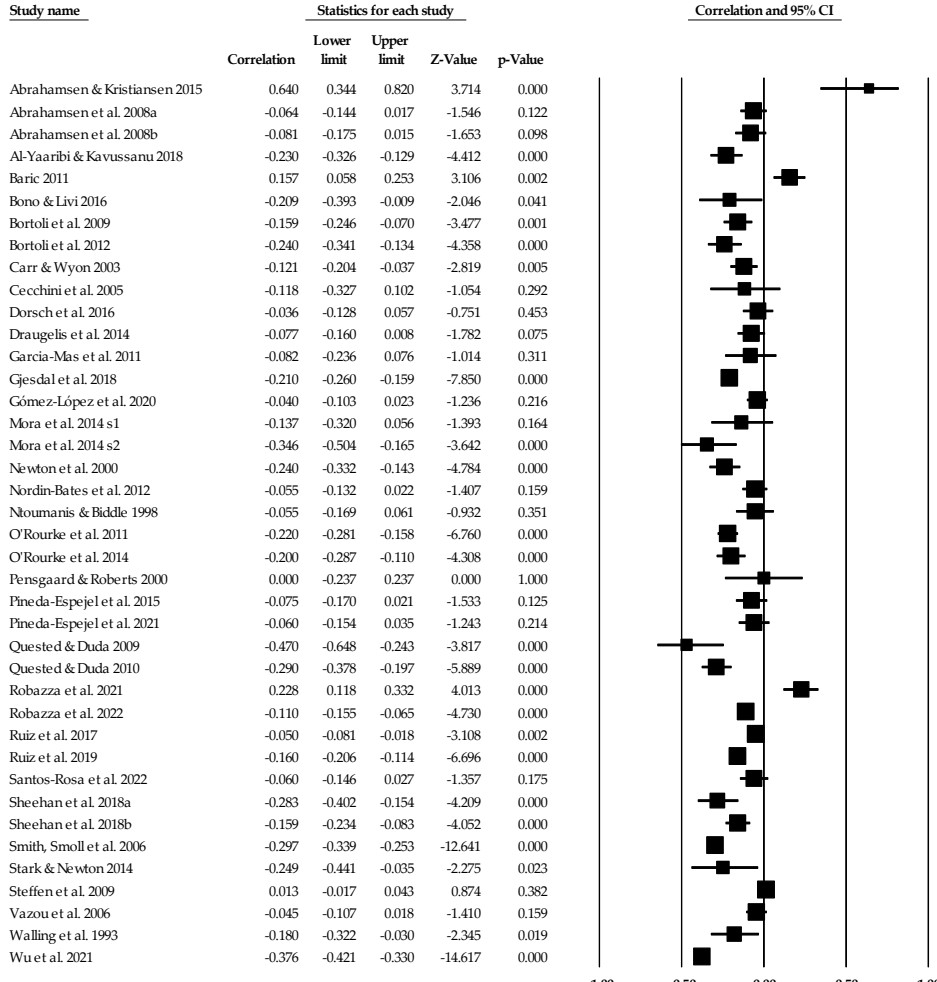

| Study name | Correlation | Lower limit | Upper limit | Z-Value | p-Value |
|---|---|---|---|---|---|
| Abrahamsen & Kristiansen 2015 | 0.640 | 0.344 | 0.820 | 3.714 | 0.000 |
| Abrahamsen et al. 2008a | -0.064 | -0.144 | 0.017 | -1.546 | 0.122 |
| Abrahamsen et al. 2008b | -0.081 | -0.175 | 0.015 | -1.653 | 0.098 |
| Al-Yaaribi & Kavussanu 2018 | -0.230 | -0.326 | -0.129 | -4.412 | 0.000 |
| Baric 2011 | 0.157 | 0.058 | 0.253 | 3.106 | 0.002 |
| Bono & Livi 2016 | -0.209 | -0.393 | -0.009 | -2.046 | 0.041 |
| Bortoli et al. 2009 | -0.159 | -0.246 | -0.070 | -3.477 | 0.001 |
| Bortoli et al. 2012 | -0.240 | -0.341 | -0.134 | -4.358 | 0.000 |
| Carr & Wyon 2003 | -0.121 | -0.204 | -0.037 | -2.819 | 0.005 |
| Cecchini et al. 2005 | -0.118 | -0.327 | 0.102 | -1.054 | 0.292 |
| Dorsch et al. 2016 | -0.036 | -0.128 | 0.057 | -0.751 | 0.453 |
| Draugelis et al. 2014 | -0.077 | -0.160 | 0.008 | -1.782 | 0.075 |
| Garcia-Mas et al. 2011 | -0.082 | -0.236 | 0.076 | -1.014 | 0.311 |
| Gjesdal et al. 2018 | -0.210 | -0.260 | -0.159 | -7.850 | 0.000 |
| Gómez-López et al. 2020 | -0.040 | -0.103 | 0.023 | -1.236 | 0.216 |
| Mora et al. 2014 s1 | -0.137 | -0.320 | 0.056 | -1.393 | 0.164 |
| Mora et al. 2014 s2 | -0.346 | -0.504 | -0.165 | -3.642 | 0.000 |
| Newton et al. 2000 | -0.240 | -0.332 | -0.143 | -4.784 | 0.000 |
| Nordin-Bates et al. 2012 | -0.055 | -0.132 | 0.022 | -1.407 | 0.159 |
| Ntoumanis & Biddle 1998 | -0.055 | -0.169 | 0.061 | -0.932 | 0.351 |
| O'Rourke et al. 2011 | -0.220 | -0.281 | -0.158 | -6.760 | 0.000 |
| O'Rourke et al. 2014 | -0.200 | -0.287 | -0.110 | -4.308 | 0.000 |
| Pensgaard & Roberts 2000 | 0.000 | -0.237 | 0.237 | 0.000 | 1.000 |
| Pineda-Espejel et al. 2015 | -0.075 | -0.170 | 0.021 | -1.533 | 0.125 |
| Pineda-Espejel et al. 2021 | -0.060 | -0.154 | 0.035 | -1.243 | 0.214 |
| Quested & Duda 2009 | -0.470 | -0.648 | -0.243 | -3.817 | 0.000 |
| Quested & Duda 2010 | -0.290 | -0.378 | -0.197 | -5.889 | 0.000 |
| Robazza et al. 2021 | 0.228 | 0.118 | 0.332 | 4.013 | 0.000 |
| Robazza et al. 2022 | -0.110 | -0.155 | -0.065 | -4.730 | 0.000 |
| Ruiz et al. 2017 | -0.050 | -0.081 | -0.018 | -3.108 | 0.002 |
| Ruiz et al. 2019 | -0.160 | -0.206 | -0.114 | -6.696 | 0.000 |
| Santos-Rosa et al. 2022 | -0.060 | -0.146 | 0.027 | -1.357 | 0.175 |
| Sheehan et al. 2018a | -0.283 | -0.402 | -0.154 | -4.209 | 0.000 |
| Sheehan et al. 2018b | -0.159 | -0.234 | -0.083 | -4.052 | 0.000 |
| Smith, Smoll et al. 2006 | -0.297 | -0.339 | -0.253 | -12.641 | 0.000 |
| Stark & Newton 2014 | -0.249 | -0.441 | -0.035 | -2.275 | 0.023 |
| Steffen et al. 2009 | 0.013 | -0.017 | 0.043 | 0.874 | 0.382 |
| Vazou et al. 2006 | -0.045 | -0.107 | 0.018 | -1.410 | 0.159 |
| Walling et al. 1993 | -0.180 | -0.322 | -0.030 | -2.345 | 0.019 |
| Wu et al. 2021 | -0.376 | -0.421 | -0.330 | -14.617 | 0.000 |

**Figure 7.** Task climate and negative affect statistics expressed as correlations (*r*) with corresponding forest plots. Figure references [24,26,41,42,44,46,48–50,56,57,60,61,63,64,69–71,78,80,81,84,86,87,93,94, 96,98,101–103,107,109,110,112,116,117,120].

**Funnel Plot of Standard Error by Fisher's Z**

**Figure 8.** Task climate and negative affect random effects plot trimmed and filled. The open circles are the data points, and the filled circles are the result of the trim and fill analysis. The clear rhombus is the mean effect size, and the filled rhombus is the trim and filled mean effect size.

*3.3. Ego Climate Individual Study Data, Synthesis of Results, and Risk of Bias across Studies*

Individual study data with corresponding forest plots for the ego climate analyses are located in Figures 9–14. Table 6 contains all the summary data for the ego climate analyses. For both positive affect and satisfaction, the random effect sizes were small in magnitude. For both sets of measures, the 95% confidence intervals remained just inside 0. However, the true prediction intervals crossed zero. Heterogeneity was present for both sets of measures. For positive affect, Orwin's *n* was 0 as this analysis utilizes the fixed-effect *r* (−0.08). The trim and fill analysis suggested that the ego climate and positive affect relationship needed correction, but the overall relationship changed only from −0.11 to −0.09 (see Figure 10). The ego climate and satisfaction relationship appeared to be influenced by Bekiari and Syrmpas [88] (see individual study data in Figures 11 and 12 for the funnel plot) in that the publication bias statistic adjusted from −0.18 to −0.30. Removal of Bekiari and Syrmpas resulted in no trim and fill adjustment and a resultant random effects correlation of −0.11.

**Table 6.** Ego/performance climate and hedonic well-being results.

| Correlate | k | r | 95% CI | 95% PI | Q | $\tau^2$ | $I^2$ | FS | Orwin | Trim/Fill | r [95% CI] |
|---|---|---|---|---|---|---|---|---|---|---|---|
| | | **Effect Size Statistics** | | | | **Heterogeneity Statistics** | | | **Bias Statistics** | | |
| PA | 40 | −0.11 | −0.16, −0.07 | −0.38, 0.17 | 503.64 | 0.02 | 92.25 | 2938 | 0 | 11R | −0.05 [−0.10, −0.00] |
| NA | 38 | 0.19 | 0.16, 0.24 | −0.05, 0.42 | 418.65 | 0.02 | 91.17 | 2063 | 39 | 8L | 0.15 [0.11, 0.20] |
| Satisfaction | 18 | −0.18 | −0.32, −0.03 | −0.70, 0.46 | 610.75 | 0.10 | 97.21 | 1311 | 16 | 7L | −0.30 [−0.43, −0.17] |
| | 17 [A] | −0.11 | −0.18, −0.04 | −0.39, 0.18 | 116.70 | 0.02 | 86.29 | 436 | 4 | 0 | No change |

Abbreviations: PA = positive affect constructs, NA = negative affect constructs, k = number of samples, CI = confidence interval, PI = prediction interval, Q = Q total between statistics, $\tau^2$ = tau-squared, $I^2$ = ratio of excess dispersion to total dispersion, FS = fail-safe number. Superscript: [A] = Bekiari and Syrmpas [88] data point removed.

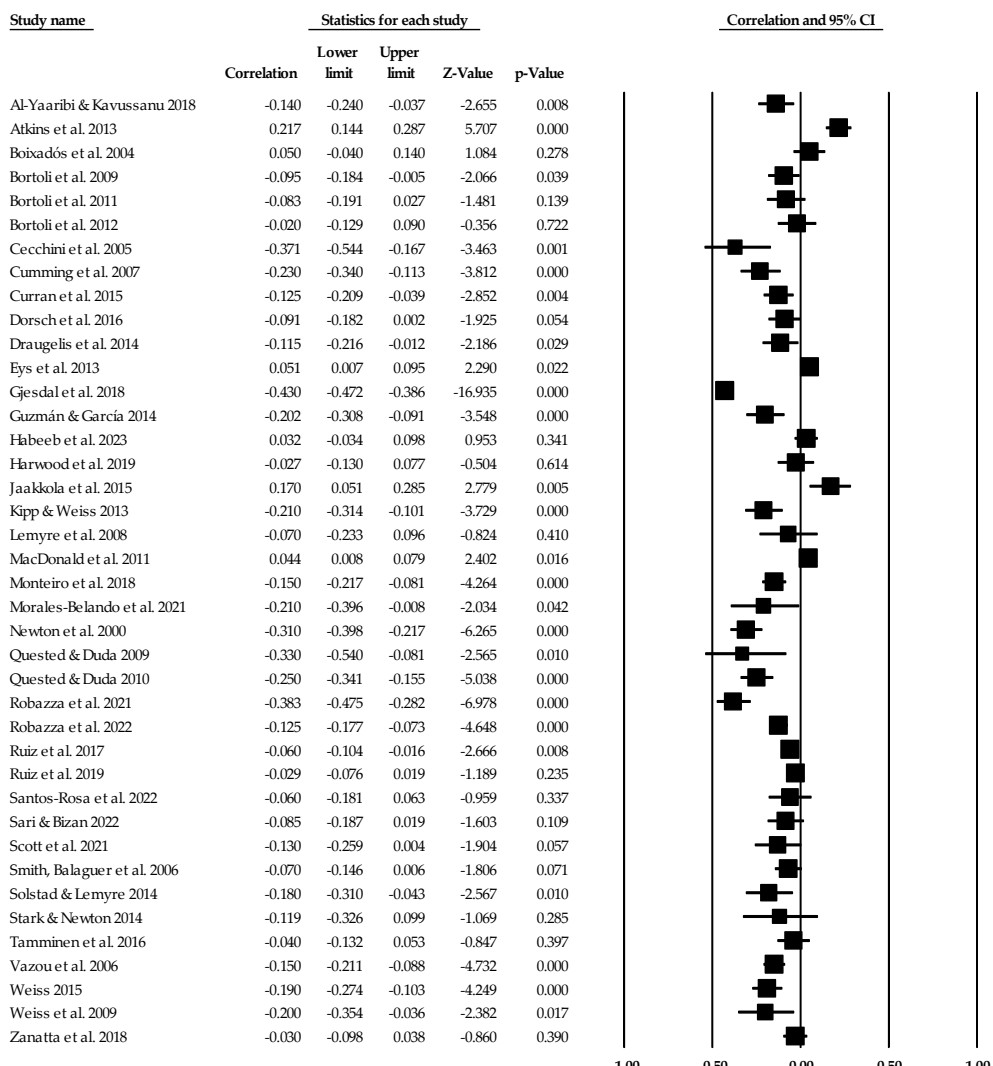

**Figure 9.** Ego climate and positive affect statistics expressed as correlations (*r*) with corresponding forest plots. Figure references [24,47–50,52,54,56,59–61,65,68,70,74–76,78,81–83,85,89,91,94–99,103, 104,111–113,115–117,119,120].

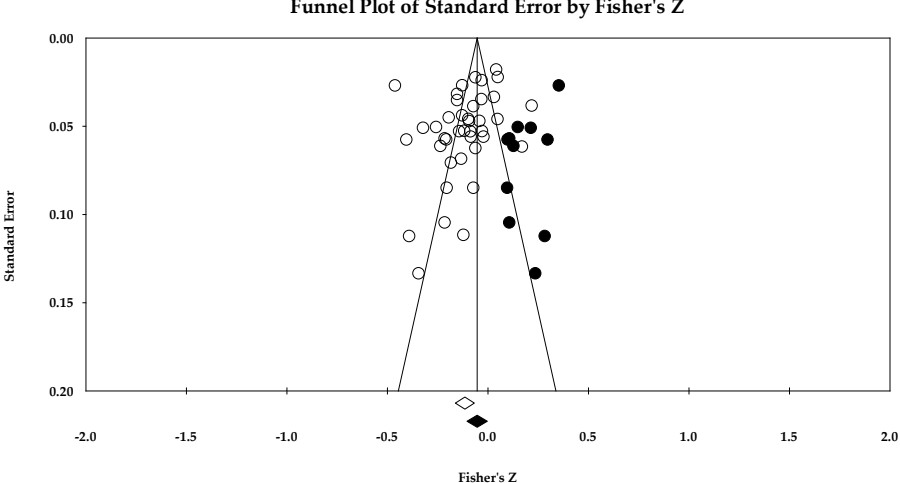

**Figure 10.** Ego climate and positive affect random effects plot trimmed and filled. The open circles are the data points. The clear rhombus is the mean effect size, and the filled rhombus is the trim and filled mean effect size.

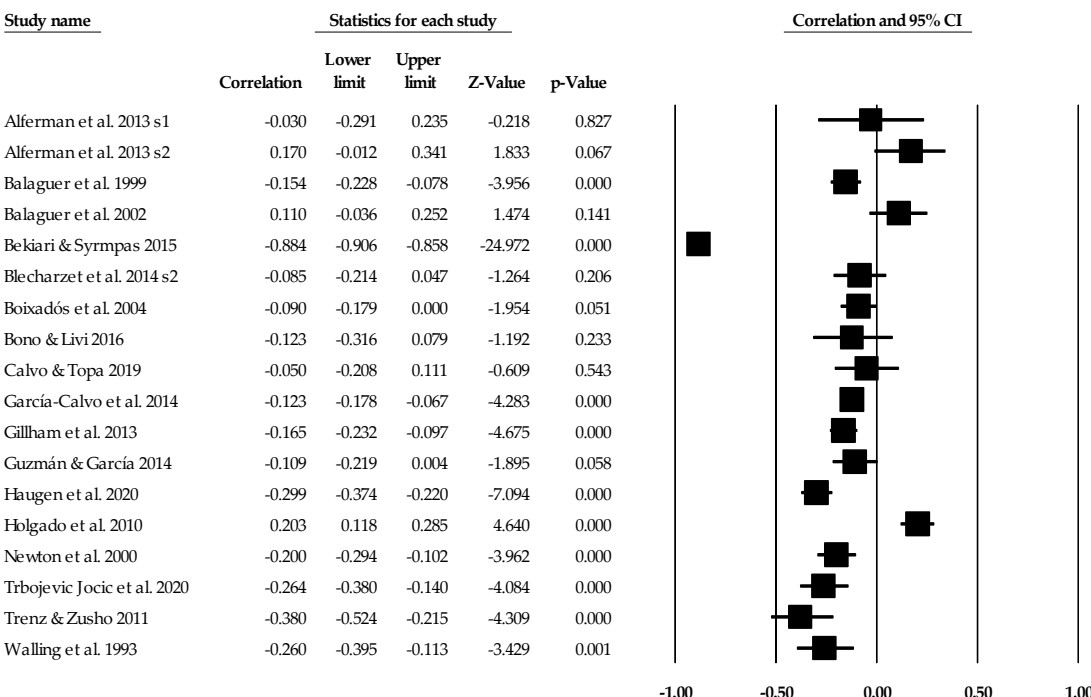

| Study name | Statistics for each study | | | | | Correlation and 95% CI |
|---|---|---|---|---|---|---|
| | Correlation | Lower limit | Upper limit | Z-Value | p-Value | |
| Alferman et al. 2013 s1 | -0.030 | -0.291 | 0.235 | -0.218 | 0.827 | |
| Alferman et al. 2013 s2 | 0.170 | -0.012 | 0.341 | 1.833 | 0.067 | |
| Balaguer et al. 1999 | -0.154 | -0.228 | -0.078 | -3.956 | 0.000 | |
| Balaguer et al. 2002 | 0.110 | -0.036 | 0.252 | 1.474 | 0.141 | |
| Bekiari & Syrmpas 2015 | -0.884 | -0.906 | -0.858 | -24.972 | 0.000 | |
| Blecharzet et al. 2014 s2 | -0.085 | -0.214 | 0.047 | -1.264 | 0.206 | |
| Boixadós et al. 2004 | -0.090 | -0.179 | 0.000 | -1.954 | 0.051 | |
| Bono & Livi 2016 | -0.123 | -0.316 | 0.079 | -1.192 | 0.233 | |
| Calvo & Topa 2019 | -0.050 | -0.208 | 0.111 | -0.609 | 0.543 | |
| García-Calvo et al. 2014 | -0.123 | -0.178 | -0.067 | -4.283 | 0.000 | |
| Gillham et al. 2013 | -0.165 | -0.232 | -0.097 | -4.675 | 0.000 | |
| Guzmán & García 2014 | -0.109 | -0.219 | 0.004 | -1.895 | 0.058 | |
| Haugen et al. 2020 | -0.299 | -0.374 | -0.220 | -7.094 | 0.000 | |
| Holgado et al. 2010 | 0.203 | 0.118 | 0.285 | 4.640 | 0.000 | |
| Newton et al. 2000 | -0.200 | -0.294 | -0.102 | -3.962 | 0.000 | |
| Trbojevic Jocic et al. 2020 | -0.264 | -0.380 | -0.140 | -4.084 | 0.000 | |
| Trenz & Zusho 2011 | -0.380 | -0.524 | -0.215 | -4.309 | 0.000 | |
| Walling et al. 1993 | -0.260 | -0.395 | -0.113 | -3.429 | 0.001 | |

**Figure 11.** Ego climate and satisfaction statistics expressed as correlations (*r*) with corresponding forest plots. Figure references [24,41,43,45,47,62,67,72,73,77,79,82,88,93,105,106,108].

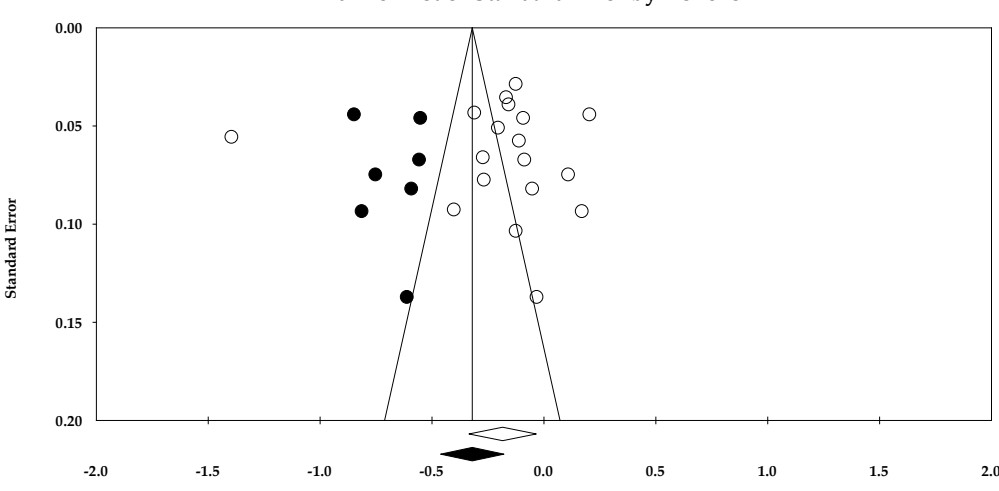

**Figure 12.** Ego climate and satisfaction random effects plot trimmed and filled. The open circles are the data points, and the filled circles are the result of the trim and fill analysis. The clear rhombus is the mean effect size, and the filled rhombus is the trim and filled mean effect size.

The ego climate and negative affect relationship like the positive affect and satisfaction relationships was small in magnitude. Unlike the other relationships, the ego climate and negative affect 95% confidence interval did not cross zero and remained small in magnitude while the true prediction interval crossed zero. As with the positive affect and satisfaction analyses, heterogeneity was present. The trim and fill analysis suggested eight missing samples though the effect size changed only to 0.15 from 0.19. Last, the bias statistics suggested that this relationship requires many studies for the relationship to change, confirming a fairly robust relationship.

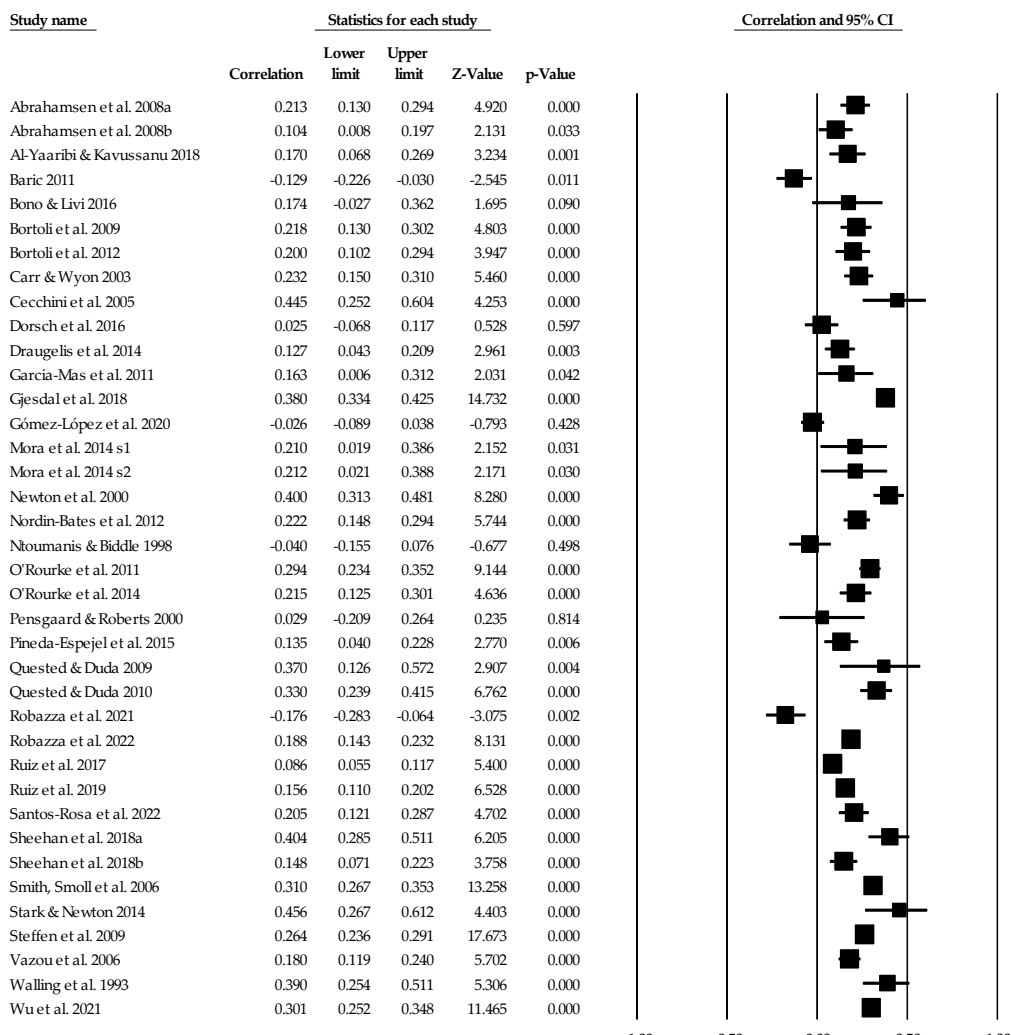

| Study name | | | | | Correlation and 95% CI |
| --- | --- | --- | --- | --- | --- |
| | Correlation | Lower limit | Upper limit | Z-Value | p-Value |
| Abrahamsen et al. 2008a | 0.213 | 0.130 | 0.294 | 4.920 | 0.000 |
| Abrahamsen et al. 2008b | 0.104 | 0.008 | 0.197 | 2.131 | 0.033 |
| Al-Yaaribi & Kavussanu 2018 | 0.170 | 0.068 | 0.269 | 3.234 | 0.001 |
| Baric 2011 | -0.129 | -0.226 | -0.030 | -2.545 | 0.011 |
| Bono & Livi 2016 | 0.174 | -0.027 | 0.362 | 1.695 | 0.090 |
| Bortoli et al. 2009 | 0.218 | 0.130 | 0.302 | 4.803 | 0.000 |
| Bortoli et al. 2012 | 0.200 | 0.102 | 0.294 | 3.947 | 0.000 |
| Carr & Wyon 2003 | 0.232 | 0.150 | 0.310 | 5.460 | 0.000 |
| Cecchini et al. 2005 | 0.445 | 0.252 | 0.604 | 4.253 | 0.000 |
| Dorsch et al. 2016 | 0.025 | -0.068 | 0.117 | 0.528 | 0.597 |
| Draugelis et al. 2014 | 0.127 | 0.043 | 0.209 | 2.961 | 0.003 |
| Garcia-Mas et al. 2011 | 0.163 | 0.006 | 0.312 | 2.031 | 0.042 |
| Gjesdal et al. 2018 | 0.380 | 0.334 | 0.425 | 14.732 | 0.000 |
| Gómez-López et al. 2020 | -0.026 | -0.089 | 0.038 | -0.793 | 0.428 |
| Mora et al. 2014 s1 | 0.210 | 0.019 | 0.386 | 2.152 | 0.031 |
| Mora et al. 2014 s2 | 0.212 | 0.021 | 0.388 | 2.171 | 0.030 |
| Newton et al. 2000 | 0.400 | 0.313 | 0.481 | 8.280 | 0.000 |
| Nordin-Bates et al. 2012 | 0.222 | 0.148 | 0.294 | 5.744 | 0.000 |
| Ntoumanis & Biddle 1998 | -0.040 | -0.155 | 0.076 | -0.677 | 0.498 |
| O'Rourke et al. 2011 | 0.294 | 0.234 | 0.352 | 9.144 | 0.000 |
| O'Rourke et al. 2014 | 0.215 | 0.125 | 0.301 | 4.636 | 0.000 |
| Pensgaard & Roberts 2000 | 0.029 | -0.209 | 0.264 | 0.235 | 0.814 |
| Pineda-Espejel et al. 2015 | 0.135 | 0.040 | 0.228 | 2.770 | 0.006 |
| Quested & Duda 2009 | 0.370 | 0.126 | 0.572 | 2.907 | 0.004 |
| Quested & Duda 2010 | 0.330 | 0.239 | 0.415 | 6.762 | 0.000 |
| Robazza et al. 2021 | -0.176 | -0.283 | -0.064 | -3.075 | 0.002 |
| Robazza et al. 2022 | 0.188 | 0.143 | 0.232 | 8.131 | 0.000 |
| Ruiz et al. 2017 | 0.086 | 0.055 | 0.117 | 5.400 | 0.000 |
| Ruiz et al. 2019 | 0.156 | 0.110 | 0.202 | 6.528 | 0.000 |
| Santos-Rosa et al. 2022 | 0.205 | 0.121 | 0.287 | 4.702 | 0.000 |
| Sheehan et al. 2018a | 0.404 | 0.285 | 0.511 | 6.205 | 0.000 |
| Sheehan et al. 2018b | 0.148 | 0.071 | 0.223 | 3.758 | 0.000 |
| Smith, Smoll et al. 2006 | 0.310 | 0.267 | 0.353 | 13.258 | 0.000 |
| Stark & Newton 2014 | 0.456 | 0.267 | 0.612 | 4.403 | 0.000 |
| Steffen et al. 2009 | 0.264 | 0.236 | 0.291 | 17.673 | 0.000 |
| Vazou et al. 2006 | 0.180 | 0.119 | 0.240 | 5.702 | 0.000 |
| Walling et al. 1993 | 0.390 | 0.254 | 0.511 | 5.306 | 0.000 |
| Wu et al. 2021 | 0.301 | 0.252 | 0.348 | 11.465 | 0.000 |

**Figure 13.** Ego climate and negative affect statistics expressed as correlations (*r*) with corresponding forest plots. Figure references [24,41,42,44,46,48,49,51,53,56,57,60,61,63,64,69–71,78,80,81,84,86,93,94, 96,98,101–103,107,109,112,116,117,120].

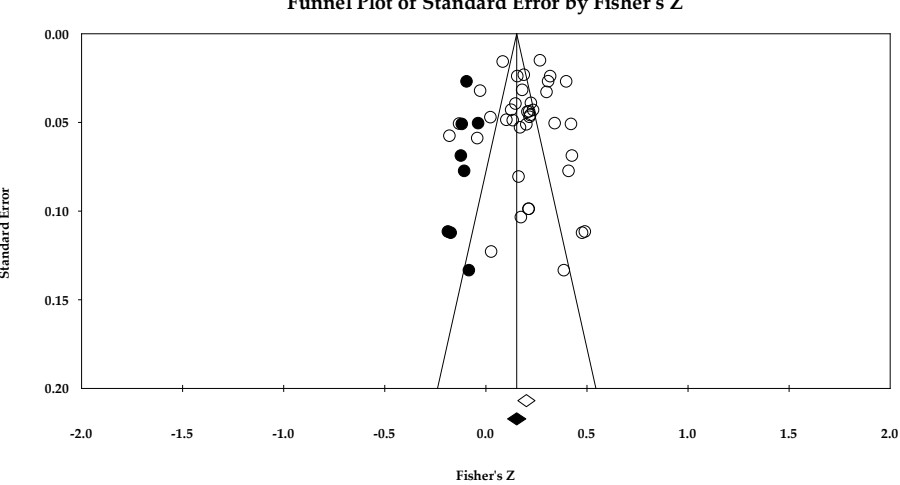

**Figure 14.** Ego climate and negative affect random effects plot trimmed and filled. The open circles are the data points, and the filled circles are the result of the trim and fill analysis. The clear rhombus is the mean effect size, and the filled rhombus is the trim and filled mean effect size.

*3.4. Additional Sensitivity Analyses*

The remove-one study analysis forest plots are located in the Supplemental File. The remove-one study analysis gauges the impact of each included study. For the task climate remove-one study analyses, the individual point estimates for each correlate category appeared to be consistent as the range of point estimates varied little even with the Abrahamsen and Kristiansen [87] data. At best, the Bekiari and Syrmpas [88] data slightly impacted the task climate and satisfaction remove-one study results. For the ego climate remove-one study analyses, the analyses for positive affect and negative affect varied little. The Bekiari and Syrmpas [88] data point influenced the satisfaction analysis to a degree. Concerning the stability of the relationships over time, also found in the Supplemental File is the cumulative analysis program by year. Minimal shifts appeared in the figures; thus, all relationships were stable across time.

*3.5. Moderator Analyses*

Meta-regression was the statistic used to evaluate the impact of percent females in each sample on the results. The percent females of each sample had no meaningful relationship with any of our motivational climate and hedonic well-being relationships as all $R^2$ values were insignificant and hovered around 0.00. Likewise, the mixed-effects sport type analyses were insignificant as the meta-analyzed correlations differed at most by 0.05 (e.g., individual $r = 0.29$ and team $r = 0.36$ for the task/mastery climate and positive affect/satisfaction analysis). However, significant differences resulted for athlete status (see Table 7). For these analyses, the positive affect and satisfaction data sets were merged so that the advanced/elite category approached a more sufficient number of samples. In both cases, the elite mean correlations were significantly less than the sub-elite correlations with both being different in the meaningfulness interpretation. The elite mean correlation for the task climate and positive affect/satisfaction was small whereas the sub-elite mean was medium in magnitude. The elite mean correlation for the ego climate and positive affect/satisfaction was negligible while the sub-elite mean correlation was small in magnitude. Unlike the relationships between the elite and sub-elite categories for the task climate and positive affect/satisfaction, the ego climate and negative affect were not significant though the pattern stayed consistent in that the mean correlations for the elite grouping were smaller than the sub-elite grouping.

**Table 7.** Athlete level and sport type moderator results.

| Relationship | Group | k | r | 95% CI | 95% PI | Q | p-Value |
|---|---|---|---|---|---|---|---|
| TC and PA, SAT | Elite | 9 | 0.23 | 0.14, 0.30 | −0.05, 0.48 | | |
| | Sub-elite | 55 | 0.34 | 0.29, 0.36 | 0.03, 0.57 | 5.66 | 0.016 |
| TC and NA | Elite | 10 | −0.09 | −0.14, −0.03 | −0.32, 0.21 | | |
| | Sub-elite | 30 | −0.14 | −0.20, −0.09 | −0.40, 0.13 | 1.81 | 0.178 |
| | Elite | 9 [A] | −0.10 | −0.15, −0.06 | −0.23, 0.02 | | |
| | Sub-elite | 30 | −0.14 | −0.20, −0.09 | −0.40, 0.13 | 1.20 | 0.273 |
| EC and PA, SAT | Elite | 8 | −0.02 | −0.11, 0.07 | −0.29, 0.26 | | |
| | Sub-elite | 47 | −0.16 | −0.21, −0.10 | −0.51, 0.25 | 6.21 | 0.013 |
| | Elite | 8 | −0.02 | −0.11, 0.07 | −0.29, 0.26 | | |
| | Sub-elite | 46 [B] | −0.13 | −0.17, −0.08 | −0.40, 0.16 | 4.49 | 0.034 |
| EC and NA | Elite | 8 | 0.16 | −0.01, 0.32 | −0.14, 0.36 | | |
| | Sub-elite | 30 | 0.21 | −0.05, 0.41 | −0.02, 0.42 | 1.63 | 0.202 |

Abbreviations: TC = task/mastery climate, PA, SAT = positive affect and satisfaction constructs, NA = negative affect constructs, EC = ego/performance climate, k = number of samples, r = mean-random-effect-modeled effect size, CI = confidence interval, PI = prediction interval, Q = Q total between statistics. Superscripts: [A] = data run without the A = Abrahamsen and Kristiansen [87] data point removed, [B] = data run without the Bekiari and Syrmpas [88] data point.

## 4. Discussion

Researchers continue to study motivation from the AGT from the original dichotomous perspective in the sport literature. The present study was a systematic review with a meta-analysis of the published literature of the task or mastery and ego or performance motivational climate and three constructs within subjective or hedonic well-being. With minimal study overlap with the Harwood and colleagues' meta-analysis (20 of 82), a focus within sport, and the examination of potential moderators, we believe that this review advances the AGT-based motivational climate literature.

### 4.1. Summary of Findings

Concerning the task climate results, our findings place a high degree of certainty that this climate is positively related to positive affect and satisfaction measures and negatively related to negative affect. For positive affect and satisfaction, both relationships resulted in medium meaningfulness correlations, whereas the negative affect correlation was small in effect size interpretation. Of interest is the task climate and positive affect effect size in this review being less than that of Harwood and colleagues [9]. In fact, the 95% confidence intervals do not overlap. Though with no sport participants, Braithwaite et al. [25] quantified task climate interventions within physical education classes. The resultant effect size for enjoyment was small. With the present data, the CMA (version 4) program provides a true effect prediction interval, which is interpreted as the range of plausible values that can include the true effect. The true predicted interval ranged from a minimal effect to a large effect. With all the information and past meta-analyses, the task climate as hypothesized since inception has no downside with positive affect and mood measures. The same conclusion can be drawn with self-rated satisfaction measures and task climate perceptions. Our satisfaction data seem to be unique to the literature and thus of great importance. Why athlete level moderated the task climate and positive affect/satisfaction relationship is unknown and open to speculation. Further down in our discussion, we propose more elite athlete research as a future direction.

As with Harwood et al. [9], the task climate and negative affect relationship was small. The ego climate relationships were all small in meaningfulness interpretation. The ego climate relationships provided a confirmation of the small relationships with positive and negative affect that Harwood and colleagues [9] reported. With negative affect, the true prediction interval provides certainty that the effect size falls between no relationship to a medium relationship. The two other quantified relationships, positive affect and satisfaction, had wider true prediction interval ranges from positive to negative values, thus casting doubt on the true effect size. As with the task climate and positive affect/satisfaction measures, the athlete level moderated the ego climate and positive affect/satisfaction relationship. This moderation, even with the one outlier removed, resulted in the elite athlete category, comprised of elite and advanced/elite athletes, resulting in a negligible correlation. The relationship with the sub-elite samples was small and negative, but even a small negative relationship to desired states lessens the potential joy of sport participation at any level.

### 4.2. Strengths, Limitations, Future Directions, and Applications

The strengths of our meta-analysis were the inclusion of 62 articles beyond the Harwood et al. [9] quantitative review with an extensive search strategy, following the PRISMA guidelines, the inclusion of satisfaction as a correlate, reporting the true prediction interval statistic provided by CMA version 4, and the examination of longstanding proposed AGT moderators. Our search resulted in a number of positive affect, negative affect, and satisfaction measures. This is also a strength. Example measures included the PANAS [121], the vigor and enthusiasm subscales from the Athlete Engagement Questionnaire [122], the pleasant and unpleasant subscales and related moods from emotional state questionnaires [123], the Sport Anxiety Scales [51], the Sport Satisfaction Scale from Duda and Nicholls joint education and sport publication [20], and the enjoyment subscale from Scan-

lan's original and updated Sport Commitment Questionnaire [124,125]. We limited each sample to only one effect size per task and ego climate analysis for each of the hedonic categories, which is another strength. For example, within the CMA program, the reporting of the three subscales by Smith and colleagues [51] of the Sport Anxiety Scale—2 was merged as were studies reporting multiple positive affect or negative affect correlations. Despite our strengths while following the structured PRISMA approach [27] to formulate and conduct a systematic review with a meta-analysis, limitations existed, stemming from the process and information provided in the included articles.

The first limitation is the number of missed studies, as we used only English in our search. The number of motivational climate studies in non-English languages (e.g., journals) is lacking in this review. Larger research teams from different countries or at least a research team member with multiple-language expertise is required to remedy this limitation. For instance, Lochbaum and colleagues' [31] meta-analysis on the $3 \times 2$ achievement goal framework included a search in the Turkish language in an attempt to minimize the language bias [126]. Biddle and colleagues' [127] systematic review on martial arts, combat sports, and mental health is an example as they searched in six different languages. The non-English studies included in our review were retrieved as the title, abstract, or keywords were written in English and supplied with the published manuscript. To extract the relevant details of methodology and results, we used Google Translate. For a few study quality ratings, we were unable to reach confidence in the provided translation. Though providing a unique finding, our coding of athlete level is another potential limitation. We applied the coding system found in the Lochbaum et al. [33] athletic identity meta-analysis based on Kyllo and Landers [128] and Swann and colleagues' [129] coding systems. The limitation stems from within sample level study participant sections, as coding depends upon the author-provided descriptions. Research with sport samples following Swann and colleagues' system will move sports science research forward. A last limitation stems again from the studies themselves as little if any random sampling or any such sampling other than convenience. However, the standard, convenience sampling could impact the data in ways unknown, as there is not enough of such studies for a comparison.

In terms of future directions, a further examination of elite sport and hedonic well-being is important, even though access to elite sport is limited. For instance, in the dichotomous AGT research, though not a firm estimate of all the AGT literature, Lochbaum et al. [6] reported that nearly half of the 260 included studies were from youth sport and approximately 19% with elite sport participants. Whether elite sport participants are less influenced by or interested in the motivational climate is unknown. Our results only indicated that the relationships were dampened. The study of potential moderators or mediators is needed to best understand the dampened relationships. For example, Ntoumanis and Biddle [42] reported self-confidence mediated the ego climate to ego orientation to state anxiety relationship. In addition to variables such as self-confidence, the athlete's relationship with their coach is a variable needing more attention as a mediator, or moderators such as the athlete's standing within the team, playing status, or length of time with the team. Another future direction concerns athletes competing at the Masters level of sport. A surprise is that not one of the mean ages in any of our studies exceeded 26, let alone approaching the age to enter for Masters athletics of 35. Hence, research with older athletes is an undeveloped area for future motivational climate and hedonic well-being research. Last, researching hedonic or eudaimonic well-being with intentionality is a future research direction. Eudaimonic well-being unlike hedonic well-being is more disputed in terms of the key concepts needed. Readers should consider Trainor and Bundon's [14] well-being commentary to gain an understanding of eudaimonic well-being frameworks.

## 5. Conclusions

In conclusion, though limitations exist with correlational data, the knowledge gained in this review is of value. First, it is evident that AGT from the original dichotomous perspective is still popular and a stronghold in the sport environment. From the list of

included and excluded studies as found in the Supplemental File, this review provides an invaluable source of references as well as meta-analyzed relationships with measures falling within the much agreed upon definition of hedonic well-being. The need to further investigate why the motivational climate and hedonic well-being relationships are smaller in elite samples is important to both researchers and practitioners. Regardless of the dampened relationships, the most practical application of our findings is that practitioners including parents and peers should focus on promoting a task or mastery climate as there is no known downside.

**Supplementary Materials:** The following supporting information can be downloaded at: https://www.mdpi.com/article/10.3390/ejihpe14040064/s1, Supplemental Table S1. PRISMA checklist. Supplemental Table S2. Reviewed study references. Supplemental Table S3. Correlates entered for each study. Supplemental Figures: Remove-one study figures: Supplemental Figure S1. Remove-one study results for task/mastery climate and positive affect/mood, Supplemental Figure S2. Remove-one study results for task/mastery climate and negative affect/mood, Supplemental Figure S3. Remove-one study results for task/mastery climate and satisfaction, Supplemental Figure S4. Remove-one study results for ego/performance climate and positive affect/mood, Supplemental Figure S5. Remove-one study results for ego/performance climate and negative affect/mood, Supplemental Figure S6. Remove-one study results for ego/performance climate and satisfaction. Supplemental Figures: Cumulative analysis by year figures: Supplemental Figure S8. Cumulative analysis by year for task/mastery climate and positive affect/mood, Supplemental Figure D9. Cumulative analysis by year for task/mastery climate and negative affect/mood, Supplemental Figure S10. Cumulative analysis by year for task/mastery climate and satisfaction, Supplemental Figure S11. Cumulative analysis by year for ego/performance climate and positive affect/mood, Supplemental Figure S12. Cumulative analysis by year for ego/performance climate and negative affect/mood, Supplemental Figure S13. Cumulative analysis by year for ego/performance climate and satisfaction.

**Author Contributions:** Conceptualization, M.L. and C.S.; methodology, M.L. and C.S.; software use, M.L. and C.S.; data retrieval and entry, M.L. and C.S.; formal analysis, M.L. and C.S.; writing—original draft preparation, M.L. and C.S.; writing—review and editing, M.L. and C.S.; supervision, M.L.; project administration, M.L.; funding acquisition, M.L. All authors have read and agreed to the published version of the manuscript.

**Funding:** Department of Kinesiology and Sport Management and Texas Tech University TrUE supported the research by purchasing licenses for the Comprehensive Meta-Analysis version 4 software.

**Institutional Review Board Statement:** Not applicable for studies not involving humans or animals.

**Informed Consent Statement:** Not applicable for studies not involving humans.

**Data Availability Statement:** All data are contained in the article tables.

**Conflicts of Interest:** The authors declare no conflicts of interest. The funders had no role in the design of the study; in the collection, analyses, or interpretation of data; in the writing of the manuscript; or in the decision to publish the results.

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
