# Peer review of "A Systematic Review with a Meta-Analysis of the Motivational Climate and Hedonic Well-Being Constructs: The Importance of the Athlete Level"

_ejihpe, doi:10.3390/ejihpe14040064_

Round 1

Reviewer 1 Report

Comments and Suggestions for Authors

The article constitutes a significant contribution to understanding the relationships between motivational climate and hedonic well-being in the context of sports participation. The article adhered to the PRISMA guidelines, enhancing its methodological credibility.

The review encompassed 82 articles that met the inclusion criteria, involving a total of 26,378 participants across various sports disciplines and levels of sports involvement, ranging from elite athletes to amateurs. The authors analyzed data spanning from 1993 to 2023, allowing for the consideration of diverse sporting contexts and cultures.

The most notable finding from the analysis is the observation that task-based motivational climate appears to have a stronger association with positive emotions and satisfaction compared to ego or performance-oriented climates. This discovery suggests that promoting a task-based motivational climate may yield greater benefits for the well-being of sports participants.

It is also worth noting that the analysis revealed some differences depending on the level of sports proficiency. The relationships between motivational climate and hedonic well-being seem to be less pronounced among elite athletes. This is an important insight that may have significant implications for sports practice, especially in terms of supporting athletes at different levels of proficiency.

This article represents a significant contribution to understanding the relationship between motivational climate and hedonic well-being in the context of sports. Its methodological rigor and thoughtful conclusions warrant publication, which may contribute to further research development on this topic and the improvement of motivational practices in sports across different proficiency levels.

Reviewer 2 Report

Comments and Suggestions for Authors

Thank you for the opportunity to review this study, which provides a systematic review of research on the relationship between motivational climate and various components of hedonic well-being. The study highlights the importance of athlete level as a potential moderator.

Overall, the manuscript is well-written and represents an advance in knowledge about the relationship between motivational climates and different components of subjective well-being. I recommend that the manuscript be published with the following minor revisions.

In the abstract, it may be best to exclude the data provided since they are already present in the corresponding section.

Additionally, it is advisable to include 'hedonic well-being' or 'subjective well-being' in the keywords.

Introduction

It is recommended to include the main authors of the AGT after the corresponding acronym, as demonstrated at the beginning of section 1.1 (line 4, ref 10-14).

On page 4, in the first paragraph, the authors hypothesize that task climate has a positive correlation with positive affect and satisfaction, and a negative correlation with negative affect and satisfaction. It is unclear whether the relationship between task climate and satisfaction is exclusively positive, as the text suggests both positive and negative relationships. Clarification is needed.

Materials and Methods

The review followed the PRISMA guide, specifying all the steps taken in the study. The selected databases and the number of articles reviewed (82 papers in total) were adequately justified.

The term 'life satisfaction' is mentioned in search 5 (pages 5-6), while in searches 18-25 only the term 'satisfaction' is used. It is unclear whether this means that any type of satisfaction was included in the search. It is important to clarify the specific type of satisfaction being referred to, whether it is satisfaction with sport, colleagues, or other aspects. The indicator used by hedonic well-being is satisfaction with life.

Figure 1 is somewhat blurred, making it difficult to read.

The review covers 30 years and provides an important update on constructs that have been extensively researched in the field of physical activity and sport. Therefore, it is a much-needed update.

Results

Table 4 shows several errors. For example: Balaguer et al. (43) used a Spanish sample, not a Norwegian one. Newton et al. (24) did not use a Spanish sample. Balaguer et al. (45) used a Spanish sample, while Bortoli et al. (60) used an Italian sample. Many other studies should be reviewed. It is recommended to examine the countries involved in each study.
